# Distinct age-related brain activity patterns in the prefrontal cortex when increasing cognitive load: A functional near-infrared spectroscopy study

**Supreeta Ranchod[1,2], Mark Rakobowchuk[1], Claudia Gonzalez** [2]*

**1** Biology Department, Faculty of Science, Thompson Rivers University, Kamloops, British Columbia, Canada, **2** Psychology Department, Faculty of Arts, Thompson Rivers University, Kamloops, British Columbia, Canada

* cgonzalez@tru.ca

## Abstract

Researchers have long observed distinct brain activity patterns in older adults compared with younger adults that correlate with cognitive performance. Mainly, older adults tend to show over-recruitment of bilateral brain regions during lower task loads and improved performance interpreted as compensation, but not observed at higher loads. However, there are discrepancies about whether increases in activity are compensatory and whether older adults can show compensation at higher loads. Our aim was to examine age-related differences in prefrontal cortex (PFC) activity and cognitive performance using functional near-infrared spectroscopy (fNIRS) during single and dual N-back tasks. Twenty-seven young adults (18–27 years) and 31 older adults (64–84 yrs) took part in the study. We used a robust fNIRS data methodology consisting of channel and region of interest analyses. Results showed differences in performance between task load conditions and age-related differences in reaction times but no age-group effects for accuracy. Older adults exhibited more bilateral PFC activation compared with young adults across all tasks and showed increases in brain activity in high compared to low load conditions. Our findings further support previous reports showing that older adults use compensatory recruitment of additional brain regions in PFC to maintain cognitive performance but go against the notion that such compensation is not present at higher cognitive loads. Additionally, our results indicate that fNIRS is a sensitive tool that can characterize adaptive cortical changes in healthy aging.

## Introduction

The older adult population is increasing worldwide [1] and aging remains associated with cognitive decline [2, 3]. The age-related decline of cognitive abilities such as memory [4], attention [5], task-switching [6], and processing speed [7] have been well documented even in the absence of pathology [3]. Furthermore, such age-related cognitive decline is associated with

**Data Availability Statement:** The data underlying the results presented in the study are available from Dr Claudia Gonzalez at DOI 10.17605/OSF.IO/745UC.

**Funding:** C.G. received a Canada Foundation for Innovation (CFI) JELF project award (number 39698) that provided the infrastructure. URL: https://www.innovation.ca/apply-manage-awards/funding-opportunities/john-r-evans-leaders-fund The funders had no role in study design, data collection and analysis, decision to publish, or preparation of the manuscript.

**Competing interests:** The authors have declared that no competing interests exist.

anatomical and functional changes in the brain [8]. It is therefore imperative to better understand and characterize the cognitive and neural changes that occur in healthy aging.

Researchers using brain imaging techniques, mainly positron emission tomography (PET) and functional magnetic resonance imaging (fMRI), have reported distinct brain activity patterns between young and older adults that correlate with decline [9]. Specifically, older adults show a decrease in brain activity specificity [10–12]. This loss in neural distinctiveness or dedifferentiation, characterized as additional activation or attenuation of brain regions [13], has often been negatively associated with cognitive performance [10, 13, 14]. However, the role of dedifferentiation as an age-specific determinant of cognitive decline has been challenged and found to have an age-invariant correlation with performance [11, 15, 16]. Furthermore, increased brain activity in older adults is not always linked with worse cognitive performance and instead, may be compensatory [11, 17]. The neural compensation view is based on observations that older adults show increases in brain activity correlated with better cognitive performance [18, 19]. Several reports have provided evidence for such compensatory effects, indicating correlations with performance in the older but not in younger adults [20–22] or that older adults with increased brain activity perform at a similar level as the younger adults during the same task (i.e., successful performance) [2, 23]. Increases in activity in older adults indicate more activity in areas recruited by younger adults and/or additional areas not recruited by younger adults, with both processes supporting cognitive performance [2].

Notably, several researchers have reported that older adults tend to activate contralateral hemispheres that were active in younger adults when performing the same task (hemisphere asymmetry reduction in older adults or HAROLD) [19, 21, 24]. Such bilateral activation in older adults, often observed in the prefrontal cortex (PFC), has been positively correlated with performance in memory [20, 25], attention shifting [26], and inhibition [27] tasks, in accordance with compensation. On its own, however, bilateral brain activity could be interpreted as neural inefficiency or dedifferentiation. Thus, Cabeza et al. [2] suggested that for increased brain activity to be interpreted as compensatory and different from dedifferentiation (i.e., lack or loss of differentiation and cognitive decline) [18], neural inefficiency (i.e., additional, non-selective brain activity that does not contribute or is detrimental to performance) [21, 28], or maintenance (i.e., preserved brain structure and function, and no evidence of cognitive decline) [12, 29], it must i) correlate with enhanced cognitive performance, and ii) be associated with insufficiencies in the neural resources needed for the task demands [2].

A common method of examining such age-related neurocognitive changes is to manipulate task demands [2], since age-related differences in brain activity seem to depend on task difficulty [30], particularly in tasks that require PFC activation [8]. Thus, increases in task cognitive demands lead to an increase in the recruitment of neural resources to meet such demands, benefiting cognitive performance (i.e., compensation); however, such over-recruitment cannot be maintained at high loads when the limit of neural resources has been reached, leading to a breakdown in cognitive performance [2, 22, 23]. These effects have been observed across age-groups with older adults over-recruiting as compensation at lower cognitive loads, then reaching neural resource limits at higher loads, leading to poorer performance (compensation related utilization of neural circuits hypothesis or CRUNCH, [21, 23]). However, Van Ruitenbeek et al. [31] recently failed to observe brain activity attenuation in older adults at high cognitive load task levels and instead, found more brain activation using fMRI [32], similar to Blum et al. [33], who also observed task-load increases in brain activity in older adults using functional near-infrared spectroscopy (fNIRS), suggesting that older adults may be able to deploy additional neural resources at high cognitive loads. Such inconsistencies make it difficult to characterize healthy aging brain activity patterns, and particularly, the role of compensation, highlighting the need for more research.

Knowledge of the neurocognitive changes in aging has been greatly enhanced by neuroimaging methods and in recent years, fNIRS has gained traction as an effective, low cost, brain imaging technique. fNIRS is based on haemodynamic changes due to metabolic demand as a proxy for neural activity and uses light emitting optodes at near-infrared wavelengths to determine changes in oxy and deoxyhemoglobin in cortical areas [34, 35]. Past research has shown that fNIRS is sensitive to age-related differences in brain activity [33, 36, 37] and to distinct cognitive loads [38–42]. In addition, due to fNIRS being an emerging technique, there have been recent improvements from traditional analyses that show better control over type I errors [43, 44], making it an increasingly robust neuroimaging method [42]. Despite advantages over other imaging techniques (e.g., larger sample sizes), there are only a few fNIRS studies that have examined healthy aging neurocognitive patterns [45]. There is also great heterogeneity between existing fNIRS studies (e.g., optode placement), making it difficult to compare findings.

The purpose of the study was to examine and characterize differences in brain activity patterns between old and young adults using fNIRS in the bilateral PFC during tasks with increasing cognitive load. We implemented two N-back tasks which required sustained attention and working memory [46], both associated with PFC processing [47, 48]. We used a single N-back task as the low load condition and a dual N-back as the high cognitive load task, given that dual tasks are known to increase processing demands [49, 50], as well as show age-related decline in older adults [51], but are less studied within the context of aging brain patterns. We hypothesized that older adults (OA) would use additional resources in the single task as part of compensatory activation, thus showing more activity compared to young adults (YA), followed by a decrease in brain activity in the dual task and poorer performance. However, previous findings [31, 33] have challenged such attenuation effect. Thus, to provide more insight into age-related differences we compared performance and brain activity across task loads and between age-groups using robust methods of fNIRS data analyses. We performed a typical channel analysis as well as ROI analyses as an estimate of active brain areas, which can be used to compare between fNIRS studies.

## Materials and methods

### Participants

A total of 66 participants (29 YA and 37 OA) were recruited for this study (June 1st, 2022-November 1st, 2022); however, data from 4 participants did not pass the fNIRS quality check (see fNIRS analysis section) and two participants terminated the experiment early. In addition, to control for cognitive abilities in older adults, we implemented the Montreal Cognitive Assessment (MOCA) [52]. Two older adult participants were therefore excluded from all analyses due to having a low score corresponding to two standard deviations (SD) below the mean (*M)* (MOCA score $M = 26.9$; SD = 1.8; normal score range corresponds to 26–30). Our final sample then consisted of 58 participants: 27 YA (18–27 years (yrs), $M = 20.56$, $SD = 2.4$ yrs, 18 Females; 9 Males) and 31 OA (64–84 yrs, $M = 71.13$, $SD = 5.5$ yrs, 21 Females, 10 Males, final MOCA score $M = 27.1$; SD = 1.6). All participants had normal or corrected-to-normal vision, had at least 6 years of formal education (YA $M = 15.07$, $SD = 1.33$ yrs; OA $M = 15.44$, $SD = 3.39$ yrs), no known neurological disorders (e.g., stroke, brain injury, Parkinson's disease), were non-smokers, not taking attention enhancing or psychoactive drugs, and were right-handed (self-reported). Participants were recruited from the local community via posters, social media, and newspaper adds, as well as from the university, and were given either a 2% bonus credit or a $10 gift card for their participation. All participants gave written informed consent prior to any experimental procedure which was witnessed and signed by the

investigators. This study was approved by the Thompson Rivers University research ethics committee.

## Procedure

Following the MOCA assessment, participants were asked to complete two N-back tasks with increasing cognitive loads corresponding to a single visuospatial 2-back task and a dual 2-back task, which included an auditory and a visuospatial stimulus presented simultaneously [49]. In both N-back tasks, participants compared a current stimulus or stimuli to the stimulus or stimuli presented 2 trials back, which requires the active maintenance, monitoring, and updating of information [47, 48]. For the dual task, the distribution of attentional resources to simultaneous processes as well as added items to remember, monitor, and update, requires additional mental resources [49] corresponding to our higher cognitive load condition. To prevent cognitive fatigue [53], we only used two load levels: a single and a dual 2-back task. We chose a 2-back as this level has shown age-related compensatory effects whilst not being too easy or too difficult for the participants to perform. Additionally, according to a recent study by Meidenbauer et al. [42], a 2-back yields the most consistent results relative to baseline (high oxy and low deoxyhaemoglobin) and also recently, Csipo et al. [54] reported no functional differences between baseline and a 1-back; both studies used fNIRS.

Stimuli were designed using E-prime 3.0 (Psychology Software Tools Inc., PA, USA), adapted to fNIRS data collection, and presented on a computer laptop (Dell Latitude 3410, 14" HD, 1920 x 1200 resolution) positioned in front of the seated participant. Participants were required to respond to the stimulus or stimuli by pressing one of two nearby buttons (S or D) on the laptop's keyboard using their right hand. While they completed these tasks, their brain activity was measured using a continuous wave fNIRS device (Brite, Artinis Medical Systems, The Netherlands). Participants were given video and face to face, step by step instructions on the N-back tasks and were also given practice trials with feedback in each trial and as a total score, prior to each N-back task. Participants were aware that their accuracy and reaction times would be measured. The entire session took approximately 90 minutes to complete, and participants were given rest breaks in between tasks.

## fNIRS set up and data acquisition

Participants were fitted with the fNIRS head cap that housed the device (Brite, Artinis, Medical Systems, The Netherlands) equipped with 10 light-emitting optodes (sources), transmitting near-infrared light at wavelengths of 650–950 nm, and 8 detectors. The head of each participant was measured to find Cz, according to the 10–20 International System [55], and aligned with the pre-marked Cz on the head cap. All optodes were placed at an optimal 3 cm from each other, except for 2 short separation channels (SSCH, source-detector distance of 1.5 cm) used to eliminate unwanted physiological noise (e.g., scalp blood flow) [56] (Fig 1). A 2 x 12 array resulted in a total of 24 channels (CH), including the two SSCHs, across right and left prefrontal cortices (12 CH per hemisphere) (Fig 1). Prefrontal regions were chosen based on previous findings from N-back tasks using fMRI [48] and fNIRS [41, 42].

The locations of sources and detectors were digitized in reference to the vertex, inion, nasion, and preauricular landmarks using a digitizing system (Patriot, Polhemus). Fig 1 shows the exported sources, detectors, and estimated source-detector pairing (i.e., channel) locations, registered to a 3D brain template (Colin27 atlas) in the Brain AnalyZIR toolbox [44]. All fNIRS data was collected using Oxysoft (Artinis, Medical Systems, The Netherlands, version 3.2.51.4) sampling at 25 Hz. Task event markers (N-back onsets and durations) were inserted from E-prime into Oxysoft via distributed component object model (DCOM) methods. Prior to data collection

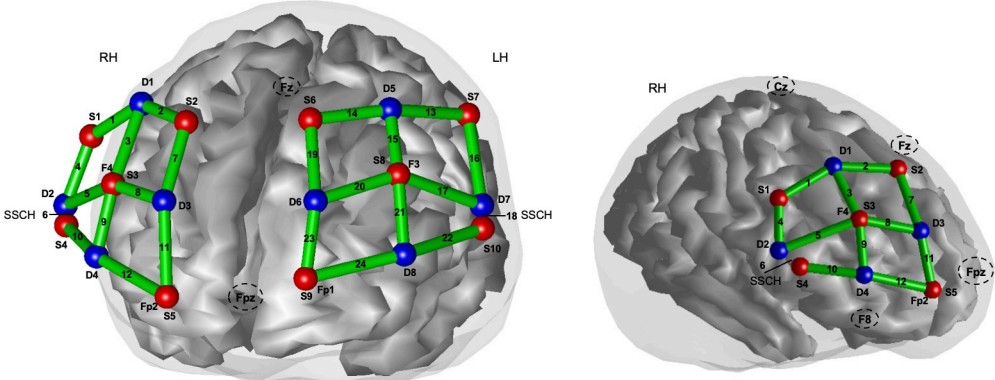

**Fig 1. Topographic organisation of the fNIRS optode array over bilateral PFC.** The image shows the 2 x 12 array with 8 detectors (D, in blue) and 10 sources (S, in red), registered to a brain template (Colin27 atlas), in a. frontal and b. lateral views. The source-detector pairings make up 24 channels (in green), 12 on the right and 12 on the left hemispheres (RH and LH, respectively). Short separation channels (SSCH) were placed in analogous locations in both hemispheres (CH 18 in LH and CH 6 in RH). Optode placements with respect to Cz, Fz, Fpz, Fp1, Fp2, F3, F4 and F8 from the 10–20 International system are identified.

and to enhance channel data quality, we performed extensive hair-clearing for all optodes, distance safe-guards were implemented to ensure that optode pairing distance remained optimal (at 3 cm), and quality control for ambient light interference was also performed.

## Single and dual N-back task

Following practice trials and once a good fNIRS signal was obtained, participants then completed two tasks with increasing load conditions consisting of a single and a dual 2-back task. For the single 2-back task, a blue box appeared at one of six possible locations (i.e., upper left, middle, and right, and lower left, middle, and right) on the screen for 0.5 s (Fig 2). Participants then had a total of 2 s to respond by pressing the "S" key when the current stimulus was the same as two trials back, or by pressing the "D" key when the stimulus was different from two trials back using their right hand (Fig 2). A new trial (same or different location) then started after the 2 s period. Similarly, for the dual 2-back task, an audible letter (i.e., G, H, K, L, P, or Q) was played for 0.5 s simultaneously with each blue box through the speakers of the laptop. Participants then indicated whether the visual, auditory, or both stimuli were the same of different from 2 trials back using the same keys as before ("S" and "D"). The audio was tested in the practice trials to make sure that the participant could hear the stimulus. All stimuli were presented in a pseudorandom order.

A 20 s rest or baseline block was presented prior to a 2-back block. During this period, participants were asked to rest while keeping their fingers on the keyboard and simply observe a white screen with the word "rest" displayed at the centre of the screen. Following each rest block, a brief "get ready!" (0.45 s) prompted the start of a 40 s single or dual 2-back block (Fig 3). There were a total of twenty N-back trials over a 40 s block in which 20% of trials were targets (i.e., a stimulus that was the same as 2 trials ago, requiring a "S" response) and 80% were non-targets (i.e., the stimulus was different from 2 trials ago, requiring a "D" response). The sequence of rest and N-back blocks were repeated 5 times in total (Fig 3). Participants performed the single task prior to the dual task [57]. Note that participants were asked to press a key (S or D) in every trial to control for motor responses in the hemodynamic response to a N-back block.

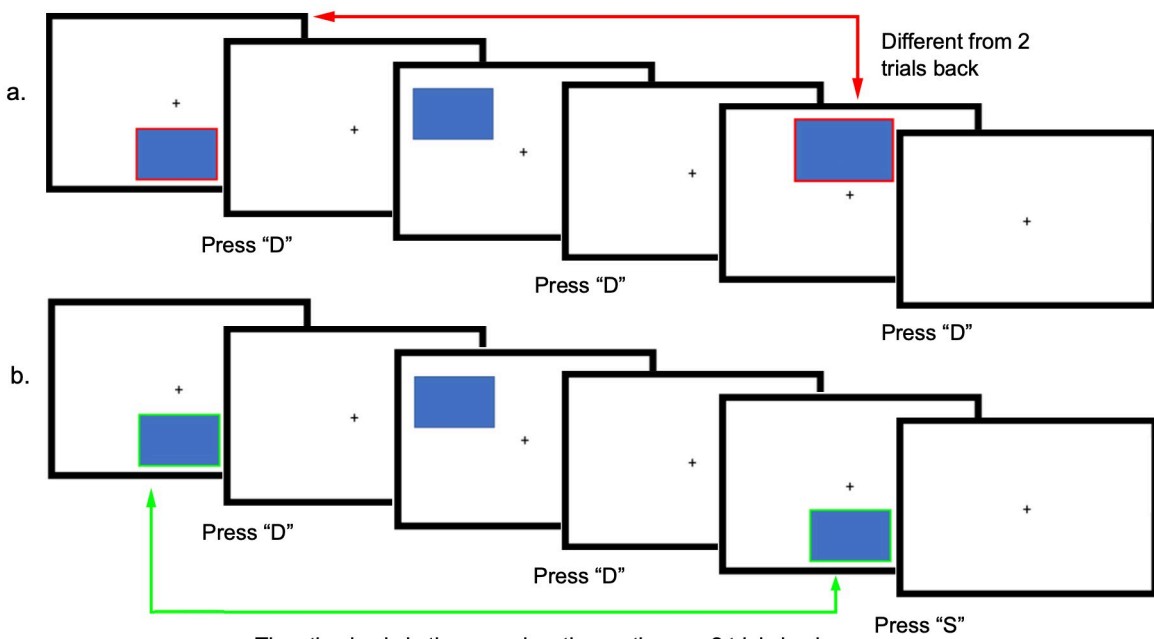

**Fig 2. Example of the single 2-back condition with visuospatial sequences.** Participants assessed whether the current box was in the same or a different location as two trials back and pressed either the "S" for same or "D" for different as two trials back in each trial. Panel a. shows a sequence in which a "D" response is required, and panel b. shows an example of a sequence in which the "S" for same key needs to be pressed. In the dual task (not shown here), the audio letter and the blue box would be presented simultaneously. The red and blue lines around the boxes are for illustration purposes and not shown to the participant.

## Analysis

### Behavioural data

Results from each participant's 2-back single and dual tasks were exported from E-prime. Trials in which no responses were observed and those in which reaction times (RTs) were < 80 ms were eliminated from further analyses to minimize predictive responses initiated ahead of the stimulus or possible guesses [58]. On average, OA had 3.25% trials removed (SD = 5.02%); whilst YA had 1.44% trials removed (SD = 2.73%). Accuracy was calculated as error rates, that is the number of incorrect responses to targets (i.e., D instead of S or *misses*) and to non-targets

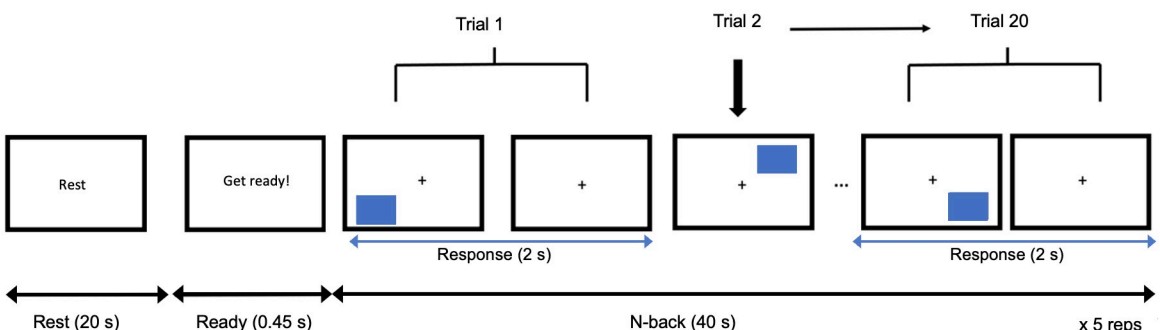

**Fig 3. Example of a single 2-back epoch for fNIRS data collection.** There was a 20 s rest followed by 40 s of a single or dual 2-back, repeated 5 times in total (x 5 reps) for both task-load conditions. Each 2-back block consisted of 20 trials.

(i.e., S instead of D *false alarms*), each divided by the total number of target and non-target trials, respectively. We also used a similar version of the discrimination index (*Pr*), which accounts for correct responses and false alarms and is therefore indicative of true positives, calculated as the rate of hits (to targets) minus the rate of false alarms [57, 59] and similar to other studies [60, 61].

RTs from correct responses were averaged for each participant in both the single and dual tasks as well as for target type (i.e., target and non-target). The statistical analysis software JASP [62] was used to conduct repeated measures analysis of variances (ANOVAs) of error rates and RTs within conditions and target types, between the two age-groups. Bonferroni corrected post-hoc tests were used to determine significant interactions. Significance level for all statistical procedures was set at $p < .05$.

### fNIRS data preprocessing and analysis

All data were exported from Oxysoft and converted in MATLAB (version 9.13.0, 2022b, Math-Works Inc., Massachusetts) for further analyses using the Brain AnalyzIR toolbox [44]. First, the raw data was converted to optical density and then converted to changes in HbO and HbR concentrations (μM) using the modified Beer-Lambert Law. An age-dependent differential path-length factor (DPF) was applied to all fNIRS data. Due to a lack of research to determine the DPF in adults older than 50, the DPF for 50 years old was used for all participants who were 50 and older [63].

Two data quality checks were implemented, one pre and one post subject-level analysis. First, we calculated the Structured Noise Index (SNI) of HbO and HbR data, as described in Zhuang et al. [53]. The SNI is an objective tool used to identify channels with high systematic noise. Similar to Zhuang et al. [53], we identified 'bad' channels as those with a SNI < 2. The total number of 'bad' channels for HbO was 11 and for HbR was 31 out of a total of 2,772 channels (number of channels x number of participants x two conditions). Therefore, participants had an average of 0.4% 'bad' channels for HbO and 1.1% of 'bad' channels for HbR. Given so few 'bad' channels, compared to Zhuang et al. [53], and the use of robust methods in the subject-level analysis which is well known to down-weight the effects of outliers (see below), no participants were excluded at this stage.

A first or subject-level general linear model (GLM) was then implemented using Brain AnalyzIR toolbox [44], with SSCHs as regressors and an autoregressive iteratively reweighted least-squares model (AR-IRLS), which includes pre-whitening and robust regression to eliminate motion and auto-correlated noise in the fNIRS signals [64, 65]. This method, including SSCH regression, has been previously shown to best control false-discovery rates due to high serial correlation errors and high-tailed noise inherent in the fNIRS signals [43]. The GLM implemented a canonical HRF, as recommended by Santosa et al. [44] for task durations > 10 s [42]. This subject-level GLM then quantified task-related haemodynamic responses resulting in beta (β) coefficient estimates in all channels for each load condition and participant.

We then implemented a group leverage analysis based on subject-level results, which identified any participant's data with significant leverage over the group's results, as described in Meidenbauer et al. [42]. This analysis led to the removal of 2 YA and 2 OA participants due to having a subject-level leverage of $p < .05$ [42]. These participants were therefore not included in the group-level analysis and their data was also removed from behavioural analyses (also see participants section).

The remaining data was used for a second or group-level analysis using a mixed effects regression model with group and task load as fixed effects and participant as a random effect. Lastly, t-test contrasts were conducted to identify significant betas (set at $p < .05$) for HbO

signals across all channels, for each task load condition, and between conditions and groups. The directions of the contrasts were selected a priori, based on previous findings showing that brain activity increases with increasing task demands, thus we assumed that the dual task would show more HbO activity vs. the single (Dual > Single), and based on OA showing more activity vs. YA (OA > YA) [2]. HbO results are typically reported over HbR, given that the HbO signals are much larger, have higher signal-to-noise ratio, are more sensitive to task-related changes [66], and are better correlated with BOLD signals in fMRI, used for comparisons [39, 67]. However, the study's raw HbR data are available online or upon request.

## fNIRS ROI analysis

The estimated locations of channels from the digitization (previously described) were further used for a region of interest (ROI) analysis. The ROI analysis uses a depth map function in the toolbox [44] which measures the distance between each estimated source-detector pairing or channel and the surface of the cortex, identifying Tailarach daemon regions of the Colin27 atlas using the nearest cortical point [42, 68]. This ROI analysis uses the spatially weighted averages of the measurement channels to test the involvement of a region in a group-level analyses. Thus, the HbO beta coefficients and the corresponding weights are incorporated into the contrast vectors. This analysis then uses a lower number of comparisons vs. channel analysis at the group-level although it can be sensitive to type II errors as described in Santosa et al. [44]. Thus, we reported both channel and ROI analyses in the present study. A total of six ROIs were identified in each hemisphere: superior frontal gyrus (Brodmann area (BA) 8), two areas in dorsolateral prefrontal cortex (DLPFC, BA 9 and BA 46), two in inferior frontal gyrus (BA 44 and BA 45); and antero-medial frontal cortex (BA 10). Fig 4 shows the coverage of ROIs

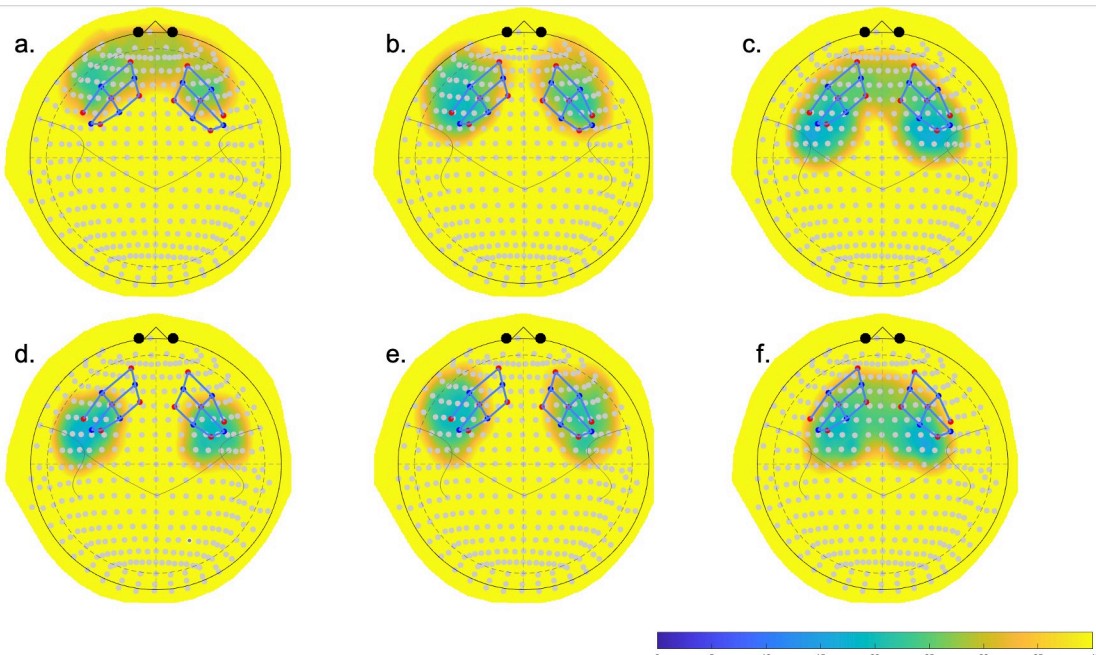

**Fig 4. ROI depth maps with the superimposed optode array over the two hemispheres.** Six ROIs were identified: a. antero-medial frontal cortex (BA 10), b. and c. dorsolateral prefrontal cortex (DLPFC) (BA 9 and BA 46, respectively), d. and e. inferior frontal gyrus (IFG) (BA 44 and BA 45, respectively); and f. superior frontal gyrus (BA 8). Channels covering blue and green areas are within range of the ROI; whilst channels over yellow and orange areas are outside of the ROI.

relative to our optode montage. Results of ROI contrasts are shown in Table 3. As with channel analysis, significant ROIs were determined at $p < .05$.

## Results

### Behavioural results

To assess whether participants were showing preference toward one type of stimulus (auditory vs. visuospatial), we performed a repeated measures ANOVA comparing error rates between the visuospatial (single and dual) and auditory stimuli. Results revealed an effect for stimulus, $F(2, 112) = 7.892$, $p < .001$, η2 = .065, indicating differences in error rates only between the single and dual visuospatial stimuli, $p < .001$, but no differences between visuospatial and auditory error rates or between groups were found, all $p > .05$, similar to Jaeggi et al. [49]. Thus, error rates to both types of stimuli in the dual task were combined.

Accuracy analysis revealed a significant interaction between task load (single and dual) and target type (target and non-target) error rates, $F(1, 56) = 5.923$, $p = .018$, η2 = .011. Post-hoc tests further revealed that participants had higher error rates to targets (i.e., misses) compared with non-targets (i.e., false alarms) in the single and dual tasks, both $p < .001$ (Fig 5). Additionally, error rates were higher in the dual task compared with the single task but only for targets, $p < .001$, and not for non-targets, $p = .58$. Error rate analysis did not reveal any error rate differences between age-groups, $p = .34$, or 3-way interactions, $p = .13$. Similarly, *Pr* (hits–false alarms) revealed an effect for load, $F(1, 56) = 26.065$, $p < .001$, η2 = .086, which suggested better accuracy in the single, $M = .65$ and $SD = .21$, compared with the dual task, $M = .53$ and $SD = .21$. No effect for group or group by load interaction was observed.

Reaction time analysis revealed a 3-way interaction between task complexity, target type, and age-group, $F(1, 56) = 5.527$, $p = .022$, η2 = .001 (Fig 6). The group effect was thus driven by this interaction. Post-hoc tests showed that OA and YA were slower at responding to targets, both $p < .001$, and non-targets, both $p < .001$, in the dual compared with the single task (Fig 6). Thus, both age-groups were slower in the dual vs. single task to all targets. Only OA showed significantly longer RTs when responding to targets compared with the non-targets in the dual task, $p < .001$, but RTs did not differ in the single task in either group, all $p > .05$. When comparing between age-groups, OA were significantly slower compared with YA when responding to targets in the single task, $p = .012$, and the dual task, $p < .001$, but were not

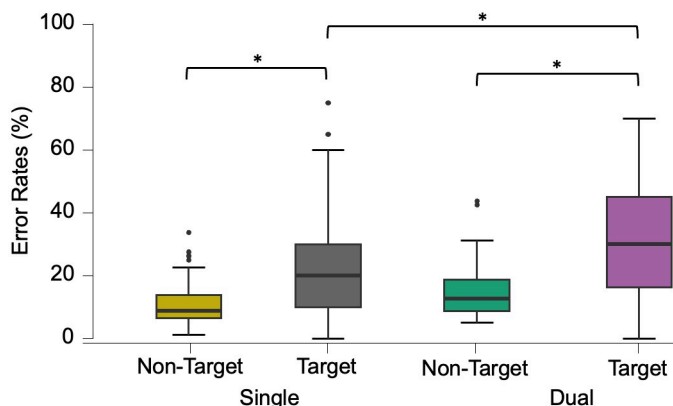

**Fig 5. Error rates (%) boxplots between task load condition and target type.** Error rates to targets correspond to misses; whilst non-target errors correspond to false alarms. The two-way interaction shows higher error rates for targets, specifically in the dual vs. single task, (*) all p < .001.

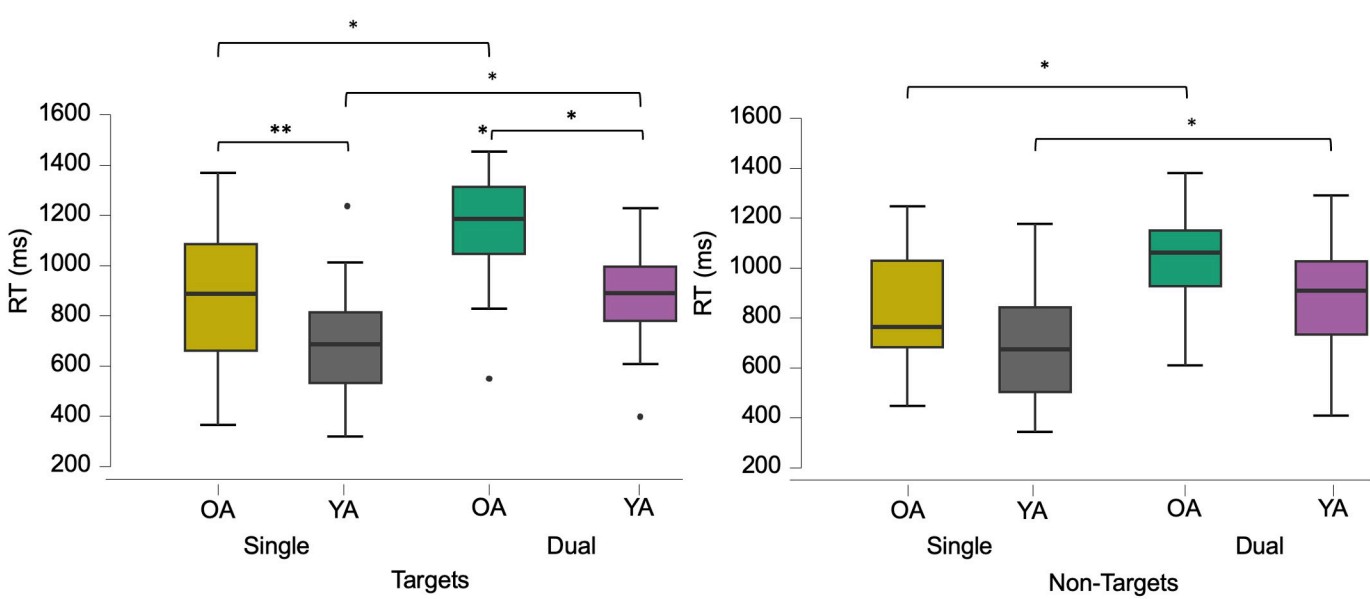

**Fig 6. Box plots for reaction times (RT) in milliseconds (ms).** The graphs show RT differences between task load conditions and groups, to targets (left) and non-targets (right). The three-way interaction shows that both YA and OA were slower to respond to targets and non-targets in the dual vs. the single task (left and right graphs), (*) $p < .001$. OA were slower vs. YA in the single, (**) $p = .012$, and dual tasks, (*) $p < .001$, but only to targets (left). In addition, OA had slower RT to targets vs. non targets in the dual task, (*) $p < .001$.

significantly different from each other when responding to non-targets ("D" response), both $p > .05$ (Fig 6).

In summary, behavioural results showed that participants had longer RTs in the dual task overall, and that OA were slower compared with YA when responding to targets. OA also showed the slowest RTs when responding to targets in the dual task ($M = 1153.03$, SD = 202.7 ms). Additionally, participants showed more error rates to targets (misses) in dual tasks compared with single tasks. However, error rates did not differ between the age-groups across task load conditions.

## fNIRS results by channel

Results from the group-level analysis are reported as channels with significant increases in HbO concentrations (i.e., calculated as β, all $p < .05$) i) relative to baseline for each load condition within each age-group (Table 1 and Fig 7), and ii) between load conditions and groups (Table 2 and Fig 8).

Relative to baseline, YA showed significant brain activity (HbO) in 2 channels on the right and 3 channels in left hemisphere, during the single task (Table 1 and Fig 7, top panel). YA also showed significant activity in 3 channels on the right and 3 channels on the left hemisphere, during the dual task. OA showed significant activity relative to baseline in channels across both hemispheres with 3 channels on RH and 7 channels on LH for the single task, and in 4 channels on RH and 12 channels on LH for the dual task (Table 1 and Fig 7, bottom panel).

Contrast analysis between load conditions (Dual > Single) revealed 3 significant channels when comparing single vs. dual brain activity in YA, with CH 14 on LH and CH 2 and 3 on RH showing significant decreases in activity in the dual relative to the single task (Table 2 and Fig 8, upper left). In contrast, OA showed mostly increases in HbO activity in CH 3 on RH, and in 3 channels (CH 15, 17, and 20) located on LH in the dual task relative to the single task.

**Table 1. Active channels relative to baseline for each load condition within each age-group.**

| Relative to baseline | Source | Detector | CH | Hemisphere | beta | T-stat | p-value |
|---|---|---|---|---|---|---|---|
| YA: Single | 2 | 1 | 2 | RH | 2.03 | 3.44 | < .001 |
| | 3 | 1 | 3 | RH | 1.62 | 3.02 | < .001 |
| | 8 | 6 | 20 | LH | 1.22 | 2.13 | .04 |
| | 8 | 8 | 21 | LH | 1.85 | 3.47 | < .001 |
| | 9 | 8 | 24 | LH | 1.55 | 2.06 | .04 |
| YA: Dual | 1 | 1 | 1 | RH | 1.18 | 2.41 | .02 |
| | 3 | 3 | 8 | RH | 1.30 | 2.49 | .01 |
| | 5 | 3 | 11 | RH | 1.42 | 2.25 | .03 |
| | 7 | 7 | 16 | LH | 1.16 | 2.36 | .02 |
| | 8 | 7 | 17 | LH | 1.09 | 2.05 | .04 |
| | 8 | 8 | 21 | LH | 1.35 | 2.44 | .02 |
| OA: Single | 2 | 1 | 2 | RH | 1.10 | 2.20 | .03 |
| | 2 | 3 | 7 | RH | 1.39 | 2.61 | .01 |
| | 3 | 3 | 8 | RH | 1.19 | 2.26 | .03 |
| | 7 | 5 | 13 | LH | 1.39 | 2.65 | .01 |
| | 6 | 5 | 14 | LH | 1.10 | 2.13 | .04 |
| | 7 | 7 | 16 | LH | 3.00 | 5.67 | < .001 |
| | 6 | 6 | 19 | LH | 1.36 | 2.41 | .02 |
| | 8 | 8 | 21 | LH | 1.48 | 2.87 | < .001 |
| | 9 | 6 | 23 | LH | 2.39 | 4.04 | < .001 |
| | 9 | 8 | 24 | LH | 1.52 | 2.43 | .02 |
| OA: Dual | 3 | 1 | 3 | RH | 1.66 | 3.34 | < .001 |
| | 3 | 2 | 5 | RH | 1.75 | 3.38 | < .001 |
| | 3 | 3 | 8 | RH | 1.39 | 2.63 | .01 |
| | 3 | 4 | 9 | RH | 1.15 | 2.3 | .02 |
| | 5 | 3 | 11 | RH | 2.94 | 4.72 | < .001 |
| | 5 | 4 | 12 | RH | 1.72 | 2.4 | .02 |
| | 7 | 5 | 13 | LH | 2.68 | 4.98 | < .001 |
| | 8 | 5 | 15 | LH | 1.82 | 3.66 | < .001 |
| | 7 | 7 | 16 | LH | 1.43 | 2.86 | < .001 |
| | 8 | 7 | 17 | LH | 1.2 | 2.22 | .03 |
| | 6 | 6 | 19 | LH | 1.18 | 2.03 | .04 |
| | 8 | 6 | 20 | LH | 2.06 | 3.76 | < .001 |
| | 8 | 8 | 21 | LH | 2.35 | 4.41 | < .001 |
| | 10 | 8 | 22 | LH | 1.69 | 2.68 | .01 |
| | 9 | 6 | 23 | LH | 2.25 | 3.81 | < .001 |
| | 9 | 8 | 24 | LH | 1.73 | 2.78 | .01 |

Significant ($p < .05$) source and detector pairings or channels (CH) across both right and left hemispheres (RH and LH, respectively).

OA also had only two channels (CH 14 and 16) on LH showing the inverse effect, that is, decreased activity in the dual task vs. the single task (Table 2 and Fig 8, upper right).

Group contrasts further revealed that during the single task, relative to YA, OA showed increased brain activity in CH 16 and 23, both channels located on LH (Table 2 and Fig 8 bottom left). Additionally, in the dual task, OA showed increased activity compared with YA across CH 13, 19, 20, and 23 located on LH (Table 2, Fig 8, bottom right). Thus, OA activated their left hemisphere more than YA.

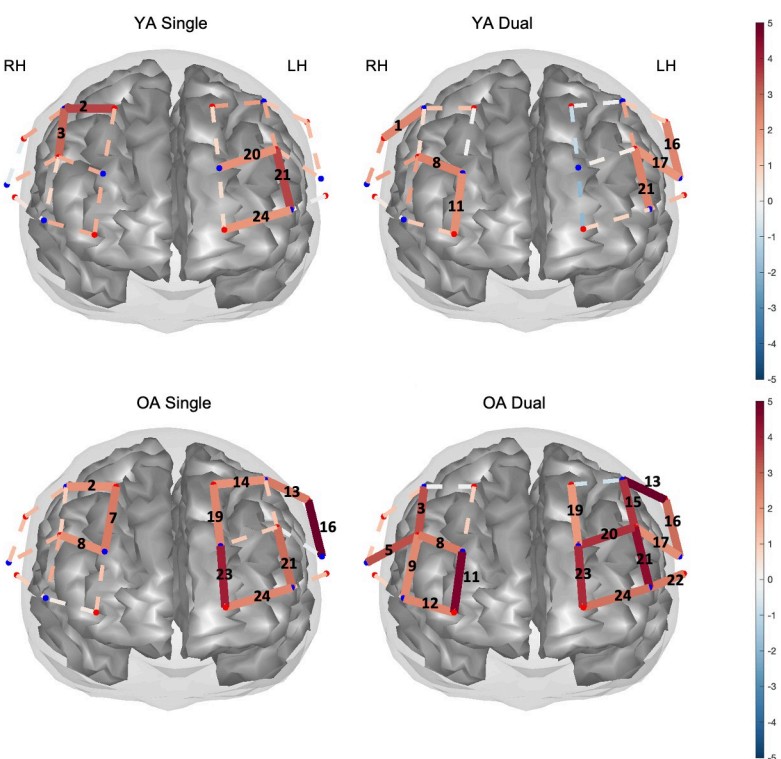

**Fig 7. Mapped T-stat values (Table 1) for HbO activity relative to baseline.** Significant ($p < .05$) channels are shown for single and dual tasks (left and right panels, respectively), for young adults (YA, upper panel) and older adults (OA, lower panel). Solid red channels correspond to significant activity or increases in HbO in a channel.

**Table 2. Group-level contrast results with significantly active channels between task load conditions and groups.**

| Contrast | Source | Detector | CH | Hemisphere | beta | T-stat | p-value |
|---|---|---|---|---|---|---|---|
| YA: Dual > Single | 2 | 1 | 2 | RH | -1.43 | -2.03 | .04 |
| | 3 | 1 | 3 | RH | -1.33 | -2.07 | .04 |
| | 6 | 5 | 14 | LH | -1.08 | -2.21 | .03 |
| OA: Dual > Single | 5 | 3 | 11 | RH | 2.23 | 2.71 | .01 |
| | 6 | 5 | 14 | LH | -1.49 | -2.25 | .03 |
| | 8 | 5 | 15 | LH | 1.25 | 2.07 | .04 |
| | 7 | 7 | 16 | LH | -1.57 | -2.38 | .02 |
| | 8 | 7 | 17 | LH | 1.39 | 2.08 | .04 |
| | 8 | 6 | 20 | LH | 1.49 | 2.13 | .04 |
| Single: OA > YA | 7 | 7 | 16 | LH | 2.33 | 3.16 | < .001 |
| | 9 | 6 | 23 | LH | 1.92 | 2.13 | .04 |
| Dual: OA > YA | 7 | 5 | 13 | LH | 2.01 | 2.42 | .02 |
| | 6 | 6 | 19 | LH | 1.68 | 2.16 | .03 |
| | 8 | 6 | 20 | LH | 1.93 | 2.40 | .02 |
| | 9 | 6 | 23 | LH | 3.43 | 4.00 | < .001 |

Significant ($p < .05$) source and detector pairings or channels (CH) across both right and left hemispheres (RH and LH, respectively) are shown.

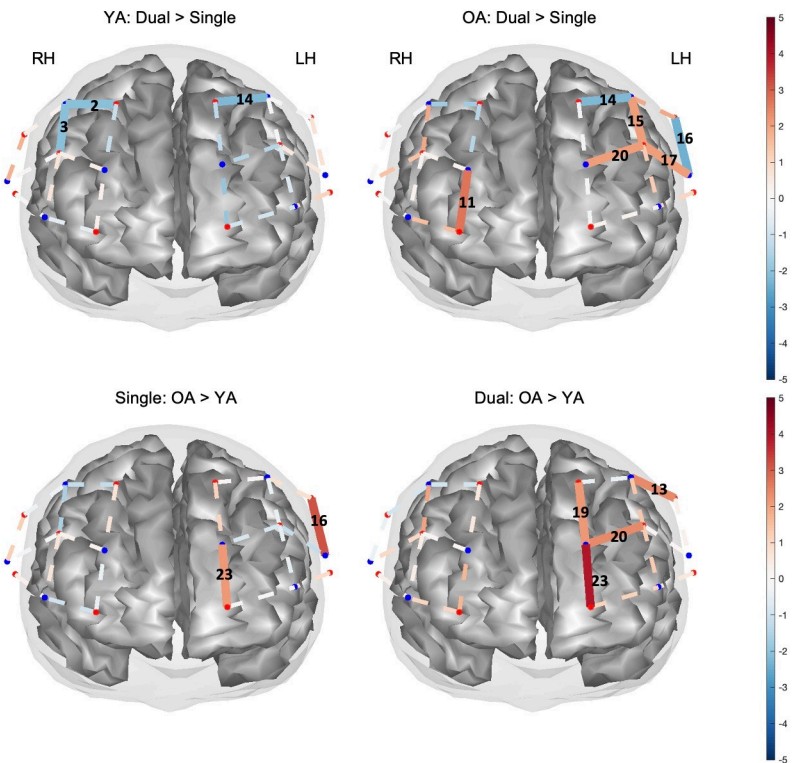

**Fig 8. Mapped T-stat values (Table 2) from group-level contrasts.** The figure shows the between load condition contrasts (Dual > Single) for each age-group (top panel) and the between group contrasts, young adults (YA) and older adults (OA), from each load condition (bottom panel). Solid red channels correspond to significant activity or increases in HbO, whilst solid blue lines correspond to significant decreases in HbO activity in a channel. Only significant channels are identified ($p < .05$).

## fNIRS results by ROIs

ROI analysis for group by load contrasts revealed that OA had more activity in the right antero-medial frontal cortex and DLPFC in the dual task relative to the single task (Table 3). YA did not show significant ROIs when contrasting task load conditions. When comparing between groups, no significantly active ROIs were found in the single task, although a trend, $p = .06$, indicated more left IFG for OA. During the dual task, OA activated the left DLPFC region more compared with YA (Table 3).

**Table 3. Group-level contrast ROIs found between task load condition and groups.**

| Contrast | Region | Beta | SD | T-stat | p-value | Power |
|---|---|---|---|---|---|---|
| YA: Single > Dual* | - | - | - | - | - | - |
| OA: Single > Dual | R Antero-medial frontal (BA 10) | 1.06 | .52 | 2.03 | .044 | .35 |
| | R DLPFC (BA 46) | 1.08 | .54 | 1.99 | .048 | .37 |
| Single: OA > YA* | - | - | - | - | - | - |
| Dual: OA > YA | L DLPFC (BA 9) | 1.58 | .40 | 3.93 | < .001 | .01 |
| | L DLPFC (BA 46) | 1.19 | .48 | 2.47 | .015 | .21 |

*Contrasts in which no significant ROIs were found, $p > .05$. DLPFC = dorsolateral prefrontal cortex; IFG = inferior frontal gyrus.

In summary, group-level analysis revealed that YA showed differential activity in only 3 channels when comparing between task load, and mostly, they showed decreased activity in the dual relative to the single task. In contrast, OA showed increases in brain activity in the high load task. Channel analysis revealed that such increases occurred in both hemispheres, however, ROI analysis revealed the right DLPFC and antero-medial as significantly active in the dual relative to the single task. It was also evident that OA had more active channels compared with YA in both load conditions across both hemispheres, however, age-group brain activity differences were observed predominantly in the LH, indicating that compared with YA, OA activated the LH more. This increase in LH activity in OA was found in DLPFC according to the ROI analysis.

## Discussion

Behavioural results revealed lower accuracy and longer RTs in the dual task compared with the single task, indicating that the dual task increased cognitive demands [57, 69]. Unexpectedly, older adults were not less accurate than the younger adult group at high loads; however, they demonstrated some behavioural differences. First, OA were slower to identify targets compared with YA who did not show delay differences between target types; second, OA were slower compared with YA when responding to these targets; and third, they were the only age-group that showed significant RT costs to targets with increased cognitive load. Slower RTs are typically associated with age-related general processing delays; however, OA were not slower to non-targets vs. YA but showed more delays in processing targets, particularly in dual situations, which indicates differential processing between the groups. After equating single task performance between OA and YA groups, Logie et al. [70] also found that OA were at a disadvantage regarding RT costs but not error rates in dual tasks, presumably due to slowing down to maintain accuracy. Speed-accuracy trade-offs have been explained in terms of OA prioritizing accuracy over speed to avoid mistakes [71] and as compensation for age-related neural inefficiencies [72, 73]. Sala-Llonch et al. [60] examined performance and brain activity (fMRI) in healthy OA and YA using a series of N-back tasks with increasing load. They showed similar behavioural performance (i.e., accuracy) between their OA high performing group and YA group in a high load 3-back task, whilst also reporting longer RTs in OA compared with YA [60]. Additionally, this lack of age-group effects on performance was associated with increased brain connectivity and BOLD responses in bilateral frontal regions in the OA [60]. Similarly, others have found comparable inhibitory performance between OA and YA [74–76]. Inhibition is a PFC function affected by age-related decline and said to hinder dual task performance [77, 78]. However, Hsieh and Lin [74] reported no age effects on performance and concluded that OA's speed-accuracy trade-off was coupled with increased frontal activation (measured as event-related potentials) to alleviate age-related deficits, similar to Hsieh and Fang [76]. These previous findings indicated that OA's 'successful' cognitive performance, defined as similar performance to YA, and/or cognitive performance trade-offs are accompanied by the recruitment of distinct brain areas compared to YA, in line with the HAROLD model and the compensation view [60, 75]. Accordingly, our results show evidence of age-related performance strategies, distinct brain activity patterns between the age-groups; specifically, more bilateral PFC activity in OA, and similar error rates between the age-groups. It is also worth noting that, though not expected, similarities in performance in this study provided an opportunity to better compare brain activity patterns between OA and YA [29].

Regarding brain activity, our main aim was to compare OA brain activity with YA across two tasks with different cognitive loads. Our hypotheses were based on research indicating that frontal brain activity increases with higher task demands. However, if capacity is exceeded,

activity decreases and OA exceed their limit more rapidly compared with YA, resulting in poorer performance compared with YA at such higher loads [19]. It is important to note that we only used two load levels, starting at an intermediate load, single 2-back vs. a 1-back [23], and 3 or 4 load levels are recommended to effectively test for CRUNCH [23, 31, 32]. Thus, we did not intend to test for CRUNCH per se but to generate a demand according to Cabeza et al.'s [2] compensation criteria. However, we expected that if OA show higher brain activation at relatively low task loads such as a 1-back and reach neural limits at lower loads compared with YA [23], that they would exhibit more brain activity vs. YA at lower loads and attenuation at higher loads. Conversely, we expected YA to show increased brain activity in the dual task, albeit it was also probable that YA would also exhibit attenuation at higher loads if the task was sufficiently demanding. Our results showed that YA had a relative decrease in brain activity in the dual task compared with the single task according to channel analyses, and ROI analyses showed no significant differences in this age group between task loads. YA may have exceeded their limits for the dual task and were not able to recruit more brain activity at this high task load condition. However, this suggests YA exceeded their limits at lower loads compared with OA, which seems unlikely.

Some researchers suggest that when the task becomes too difficult, participants simply "give up" or disengage from the task [38, 40], making it difficult to identify whether decreases in performance are due to reaching a cognitive resource limit per se, or caused by a different factor such as task fatigue, anxiety, or lack of motivation. For example, Jaeggi et al. [49] attributed YA's increased PFC activity in dual relative to single tasks to trying to succeed in the complex task despite poorer performance and suggested that decreases in activity are mainly observed when participants stop trying. Unlike Jaeggi et al. [49, 57], no post task interviews were implemented; however, such dual task disengagement would explain YA's decreased brain activity and poorer performance in the present study, whilst OA were still attempting to compensate at higher loads, resulting in comparable performance between the age-groups. Indeed, YA's accuracy (% of correct targets) was lower compared to Jaeggi et al.'s [49] 2-back single and dual tasks suggesting that our YA sample found the tasks challenging. Additionally, imaging studies have indicated that when PFC areas (e.g., DLPFC) are already activated in the single version of the task, no further increases are observed in dual tasks, in line with our ROI results from YA [49, 79].

Contrast analysis revealed that OA had mostly increases in brain activity, shown in one RH channel and 3 LH channels, in the dual relative to the single task. ROI analysis also revealed increases in right antero-medial frontal cortex and DLPFC for the high load dual task relative to the single task in OA. Additionally, ROI analyses did not show significant areas in the single but showed significant left DLPFC in OA relative to YA. The increases in brain activity at higher load conditions in the OA does not support previous findings indicating that OA reach neural limits at lower loads and exhibit attenuation of PFC brain activity relative to lower loads (e.g., 1-back vs. 3-back) [23]. However, others have found a similar effect, i.e. increases in OA's brain activity with increasing load [31–33, 76]. Jamadar's [32] fMRI study showed no or very little difference in brain activity across brain regions between OA and YA at intermediate load levels, similar to our channel and ROI analyses of the single task and showed increased activity in OA compared with YA at the highest load level, similar to our high load dual task, challenging the CRUNCH model's predictions [19]. Given our task accuracy scores vs. other studies [22, 23, 49] and YA's brain activity patterns in the higher load, OA results cannot be explained by the task not being sufficiently complex, or by our sample size, which was larger compared with Mattay et al. [23], whose age-group by load interaction did not reach significance. Instead, our results suggest that the OA group was able to recruit additional PFC brain areas with increasing demands [31, 32]. Interestingly, ROI demonstrated such increase in

activity in the right hemisphere (right DLPFC) during dual relative to the single task, thus the left hemisphere may have been similarly active between task loads. This suggests that better performance may also rely on being able to maintain right PFC activity [60, 80]. Thus, it is possible that OA were working hard and trying their best by maintaining frontal cortex sufficiently engaged, leading to similar performance between the age-groups. However, we also propose that comparable performance in the dual task was influenced by YA not responding to the additional load by increasing recruitment of neural resources and may not have worked as hard on the task. It remains unclear whether an additional load would elicit lower brain activity in OA or whether a lower level would also show higher activity relative to YA.

Differences between our findings and previous studies may also be task related as Mattay et al. [23] implemented a pure N-back rather than a dual task. A dual task may elicit other functions beyond working memory, such as attention shifting and the so-called task coordination [81], which may show distinct levels of reserve in healthy OA, though more research is needed to determine such distinct effects. However, our study provides evidence that compensatory brain-behaviour relationship is not necessarily confined to one type of task. Also, we suggest that using tasks that require minimal memory resources as in a 0 or 1-back [23, 80] may enhance differences between age-groups, not seen in our study.

An important finding from the present study was OA's greater left hemisphere activation relative to YA and overall increased bilateral PFC activity, which may have aided their performance. Recently, Seider et al. [17] found that OA with greater dedifferentiation also had better performance (i.e., compensatory dedifferentiation), similar to Sala-Llonch et al.'s [60] high performing OA who showed bilateral PFC activity and comparable performance with YA, and Cabeza et al.'s [19] high-scoring OA participants (also see [80]). Such results have been linked with higher intelligence and overall, greater cognitive reserve [19, 60]. Thus, our OA sample may have been similar to their high performing group, according to MOCA scores and years of education, and were able to activate additional brain areas to compensate for age-related decline; thus, OA were not deficient enough to show attenuation and poorer performance at higher loads. Our age-related results are therefore consistent with the HAROLD model and the compensation view, given that only OA showed such increased brain activity patterns, similar to previous reports [60, 80], suggested to increase with increasing demands (load dependent) and associated with successful performance, in line with Cabeza et al.'s [2, 19] compensation criteria.

There are some considerations regarding the present study's findings. Although our aim was to examine healthy aging brain function and behaviour, our data may be limited to a cognitively high functioning OA sample. Future research should consider including individuals with varied levels of cognitive abilities and implement a priori study designs comparing low vs. high reserve groups, possibly as a main predictor over age, and/or implement multivariate analysis to directly compare brain and behaviour measures, avoiding data averaging and minimizing comparisons [42]. Future research should also include measures of cognitive fatigue and motivation. Although results provided some evidence, we cannot argue against CRUNCH unlike Jamadar [32], who used 4 loads and also reported inverse effects. Moreover, the major difficulty lies in assessing whether such over-recruitment in bilateral PFC is indeed useful and compensatory, which the present study cannot clarify. Although a TMS study showed that OA rely more on both hemispheres compared with YA [82], another study using multivariate analysis reported that the over-recruitment of brain areas did not carry additional memory information and instead, corresponded to neural inefficiency [83]. Taken together, we cannot discard alternative explanations for such overactivation patterns in our OA group, e.g., maintenance, dedifferentiation, or neural inefficiency.

## Conclusion

We replicated findings of age-related increased brain activity in bilateral PFC. Although some researchers interpret bilateral activity as a failure to recruit specialized areas (i.e., dedifferentiation), we provided further evidence of over-recruitment in bilateral areas in OA that further increased at a higher load and was linked with successful performance. Thus, such over-recruitment in PFC was considered compensatory. Our results are inconsistent with the notion that OA show early compensation at low loads compared with YA, followed by attenuation at high loads, resulting in age-related differences in cognitive performance. However, there are several reports suggesting that OA can recruit additional resources at higher loads and report the inverse effect proposed by the CRUNCH model. Thus, more research is needed to examine such effects by implementing tasks with > 2 loads. We conclude that fNIRS is a sensitive tool to examine distinct effects of load and age-related differences in brain activity. In addition, with the emergence of novel, robust analyses tools that correct for false discoveries, fNIRS is a convenient technique that can be used to study larger samples (vs. fMRI) and examine brain activation across distinct cognitive abilities and tasks to better characterize the neuro-cognitive changes across the lifespan.

## Author Contributions

**Conceptualization:** Claudia Gonzalez.

**Data curation:** Supreeta Ranchod, Mark Rakobowchuk, Claudia Gonzalez.

**Formal analysis:** Supreeta Ranchod, Mark Rakobowchuk, Claudia Gonzalez.

**Funding acquisition:** Claudia Gonzalez.

**Investigation:** Supreeta Ranchod, Mark Rakobowchuk, Claudia Gonzalez.

**Methodology:** Claudia Gonzalez.

**Project administration:** Claudia Gonzalez.

**Resources:** Claudia Gonzalez.

**Software:** Mark Rakobowchuk, Claudia Gonzalez.

**Supervision:** Mark Rakobowchuk, Claudia Gonzalez.

**Validation:** Claudia Gonzalez.

**Visualization:** Claudia Gonzalez.

**Writing – original draft:** Supreeta Ranchod, Mark Rakobowchuk, Claudia Gonzalez.

**Writing – review & editing:** Supreeta Ranchod, Mark Rakobowchuk, Claudia Gonzalez.

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
