## [Decision Letter · Decision Letter 0]

15 Sep 2023

PONE-D-23-26564Distinct age-related brain activity patterns in the prefrontal cortex when increasing cognitive load: a functional near-infrared spectroscopy studyPLOS ONE

Dear Dr. Gonzalez,

Thank you for submitting your manuscript to PLOS ONE. After careful consideration, we feel that it has merit but does not fully meet PLOS ONE’s publication criteria as it currently stands. Therefore, we invite you to submit a revised version of the manuscript that addresses the points raised during the review process.

ACADEMIC EDITOR: Reviewers have suggested some minor changes. Authors are encouraged to revise upon reviewers' comments and submit the revised version.

We look forward to receiving your revised manuscript.

Kind regards,

Noman Naseer, PhD

Academic Editor

PLOS ONE

Journal Requirements:

Additional Editor Comments (if provided):

Reviewers have suggested some minor changes. Authors are encouraged to revise upon reviewers' comments and submit the revised version.

Reviewers' comments:

Reviewer's Responses to Questions

**Comments to the Author**

1. Is the manuscript technically sound, and do the data support the conclusions?

Reviewer #1: Yes

Reviewer #2: Yes

2. Has the statistical analysis been performed appropriately and rigorously? 

Reviewer #1: Yes

Reviewer #2: Yes

3. Have the authors made all data underlying the findings in their manuscript fully available?

Reviewer #1: Yes

Reviewer #2: Yes

4. Is the manuscript presented in an intelligible fashion and written in standard English?

Reviewer #1: Yes

Reviewer #2: Yes

5. Review Comments to the Author

Reviewer #1: The researchers used functional near-infrared spectroscopy to measure prefrontal cortex activity during single and dual N-back tasks in a group of 27 young adults and 31 older adults. The findings revealed that there were performance differences between task load conditions, and older adults exhibited slower reaction times, but both age groups showed similar levels of accuracy. Importantly, older adults displayed increased bilateral PFC activation across all tasks and even more brain activity during high-load conditions compared to low-load conditions. This study suggest that older adults continue to use compensatory recruitment of additional PFC brain regions to maintain cognitive performance, even at higher cognitive loads.

Congratulations to the authors of this study. I have thoroughly enjoyed reading this manuscript. The manuscript is formally clear, providing a well-structured and up-to-date review of relevant literature on the topic. Both the research objectives and methods are clearly articulated. The results are well-documented and effectively illustrated. The discussion comprehensively addresses key aspects arising from the study's findings.

After thoroughly reviewing the manuscript, I have only identified a few minor points related to the formatting of certain cited works.

Page 30 and reference number 31: date is May 9

Page 31 and reference number 43: please remove the duplicated page.

Reviewer #2: Summary Statement:

The present study aimed to examine age related differences in PFC activity utilizing fNIRS and cognitive performance (accuracy and RT) during 2-back visuospatial only and 2-back dual with visuospatial and audio. The authors found that older adults exhibiting more bilateral PFC activation was used as a compensatory mechanism to maintain cognitive performance. Accuracy was unaffected in OA; however, RT was slower, which may be due to OA focus on performing accurately. Overall, the results of the work is well written and organized appropriately and is of interest of the field of neurocognitive aging research. Statistical tests are appropriate. Technical aspects of the paper and scientific rigor are sufficient. Overall, this work will make a significant contribution to neuroscience aging research as the hemodynamic response in aging provides conflicting results, as described by the authors. Minimal critiques are presented below:

• Need a clear rationale for why RT that were less than 80ms were removed.

• Stated that authors used FDR corrections but then indicate that significance is set at p<0.05- this either needs to be changed to q<0.05 or mentioning FDR corrections needs to be removed.

• Tables P-values right column, the > sign should be changed to <0.001.

6. PLOS authors have the option to publish the peer review history of their article (what does this mean?). If published, this will include your full peer review and any attached files.

Reviewer #1: No

Reviewer #2: **Yes: **Cameron D. Owens

---

## [Author Response · Author response to Decision Letter 0]

18 Sep 2023

Response to reviewers

C.G. et al: We would like to thank the reviewers for their thoroughness and feedback of our manuscript. Below, we address the reviewers’ suggestions:

Reviewer #1: The researchers used functional near-infrared spectroscopy to measure prefrontal cortex activity during single and dual N-back tasks in a group of 27 young adults and 31 older adults. The findings revealed that there were performance differences between task load conditions, and older adults exhibited slower reaction times, but both age groups showed similar levels of accuracy. Importantly, older adults displayed increased bilateral PFC activation across all tasks and even more brain activity during high-load conditions compared to low-load conditions. This study suggest that older adults continue to use compensatory recruitment of additional PFC brain regions to maintain cognitive performance, even at higher cognitive loads.

Congratulations to the authors of this study. I have thoroughly enjoyed reading this manuscript. The manuscript is formally clear, providing a well-structured and up-to-date review of relevant literature on the topic. Both the research objectives and methods are clearly articulated. The results are well-documented and effectively illustrated. The discussion comprehensively addresses key aspects arising from the study's findings.

After thoroughly reviewing the manuscript, I have only identified a few minor points related to the formatting of certain cited works.

Page 30 and reference number 31: date is May 9 

C.G. et al: We appreciate the positive feedback. Thank you for spotting this. The date has been updated (see references section).

Page 31 and reference number 43: please remove the duplicated page.

C.G. et al: We have removed the duplicated page in that and two other references (see references section). 

Reviewer #2: Summary Statement:

The present study aimed to examine age related differences in PFC activity utilizing fNIRS and cognitive performance (accuracy and RT) during 2-back visuospatial only and 2-back dual with visuospatial and audio. The authors found that older adults exhibiting more bilateral PFC activation was used as a compensatory mechanism to maintain cognitive performance. Accuracy was unaffected in OA; however, RT was slower, which may be due to OA focus on performing accurately. Overall, the results of the work is well written and organized appropriately and is of interest of the field of neurocognitive aging research. Statistical tests are appropriate. Technical aspects of the paper and scientific rigor are sufficient. Overall, this work will make a significant contribution to neuroscience aging research as the hemodynamic response in aging provides conflicting results, as described by the authors. Minimal critiques are presented below:

• Need a clear rationale for why RT that were less than 80ms were removed.

C.G. et al: We have now added to this section. Reaction times < 80 ms were removed since these would correspond to guesses rather than indicating a correct or an error in working memory. These reaction times indicate that participants initiated the movement ahead of the current stimulus. We have added the rationale and a citation into the analysis section, behavioural data (page 11): “to minimize predictive responses initiated ahead of the stimulus or possible guesses (58).” 

• Stated that authors used FDR corrections but then indicate that significance is set at p<0.05- this either needs to be changed to q<0.05 or mentioning FDR corrections needs to be removed.

C.G. et al: Thank you, yes, we agree with the reviewer. We have removed FDR and kept p<0.05 as the reviewer suggested. 

• Tables P-values right column, the > sign should be changed to <0.001.

C.G. et al: Thank you for alerting us to this error. The sign has been modified for all tables (1-3).

---

## [Decision Letter · Decision Letter 1]

12 Oct 2023

Distinct age-related brain activity patterns in the prefrontal cortex when increasing cognitive load: a functional near-infrared spectroscopy study

PONE-D-23-26564R1

Dear Dr. Gonzalez,

We’re pleased to inform you that your manuscript has been judged scientifically suitable for publication and will be formally accepted for publication once it meets all outstanding technical requirements.

Kind regards,

Noman Naseer, PhD

Academic Editor

PLOS ONE

Additional Editor Comments (optional):

Associate Editor: The paper has been revised well upon reviewers' comments.

Reviewers' comments:

Reviewer's Responses to Questions

**Comments to the Author**

1. If the authors have adequately addressed your comments raised in a previous round of review and you feel that this manuscript is now acceptable for publication, you may indicate that here to bypass the “Comments to the Author” section, enter your conflict of interest statement in the “Confidential to Editor” section, and submit your "Accept" recommendation.

Reviewer #1: All comments have been addressed

Reviewer #2: All comments have been addressed

2. Is the manuscript technically sound, and do the data support the conclusions?

Reviewer #1: (No Response)

Reviewer #2: Yes

3. Has the statistical analysis been performed appropriately and rigorously? 

Reviewer #1: (No Response)

Reviewer #2: Yes

4. Have the authors made all data underlying the findings in their manuscript fully available?

Reviewer #1: (No Response)

Reviewer #2: Yes

5. Is the manuscript presented in an intelligible fashion and written in standard English?

Reviewer #1: (No Response)

Reviewer #2: Yes

6. Review Comments to the Author

Reviewer #1: I have reviewed the revised manuscript and I report that the authors have effectively addressed my previous concerns and suggestions. The manuscript has been substantially improved, and I recommend it for publication.

Reviewer #2: (No Response)

7. PLOS authors have the option to publish the peer review history of their article (what does this mean?). If published, this will include your full peer review and any attached files.

Reviewer #1: No

Reviewer #2: No

---

## [Editor Report · Acceptance letter]

20 Nov 2023

PONE-D-23-26564R1 

Distinct age-related brain activity patterns in the prefrontal cortex when increasing cognitive load: a functional near-infrared spectroscopy study 

Dear Dr. Gonzalez:

I'm pleased to inform you that your manuscript has been deemed suitable for publication in PLOS ONE. Congratulations! Your manuscript is now with our production department. 

Kind regards, 

on behalf of

Dr. Noman Naseer 

Academic Editor

PLOS ONE